# Fabrication and Characterization of Paclitaxel and Resveratrol Loaded Soluplus Polymeric Nanoparticles for Improved BBB Penetration for Glioma Management

**DOI:** 10.3390/polym13193210

**Published:** 2021-09-22

**Authors:** Talib Hussain, Sathishbabu Paranthaman, Syed Mohd Danish Rizvi, Afrasim Moin, Devegowda Vishakante Gowda, Gehad Muhammed Subaiea, Mukhtar Ansari, Abulrahman Sattam Alanazi

**Affiliations:** 1Department of Pharmacology and Toxicology, College of Pharmacy, University of Hail, Hail 81442, Saudi Arabia; mdth_ah@yahoo.com (T.H.); gehadsubaieaphd@gmail.com (G.M.S.); 2Department of Pharmaceutics, JSS College of Pharmacy, Mysuru 570015, India; sathishbabu.p94@gmail.com (S.P.); dvgowda@jssuni.edu.in (D.V.G.); 3Department of Pharmaceutics, College of Pharmacy, University of Hail, Hail 81442, Saudi Arabia; 4Department of Clinical Pharmacy, College of Pharmacy, University of Hail, Hail 81442, Saudi Arabia; mukhtaransari@hotmail.com (M.A.); aas2@hotmail.com (A.S.A.)

**Keywords:** brain cancer, combinational therapy, glioma, nano-biotechnology, polymeric nanoparticles

## Abstract

Gliomas are one of the prominent cancers of the central nervous system with limited therapeutic modalities. The present investigation evaluated the synergistic effect of paclitaxel (PAX) and resveratrol (RESV)-loaded Soluplus polymeric nanoparticles (PNPs) against glioma cell lines along with in vivo pharmacokinetics and brain distribution study. PAX-RESV-loaded PNPs were prepared by the thin film hydration technique and optimized for different dependent and independent variables by using DoE (Design-Expert) software. The in vitro physiochemical characterization of prepared PAX-RESV-loaded PNPs exhibited appropriate particle size, PDI and % encapsulation efficiency. Cytotoxicity assay revealed that PTX-RESV loaded PNPs had a synergistic antitumor efficacy against C6 glioma cells compared with single and combined pure drugs. Finally, the pharmacokinetic and brain distribution studies in mice demonstrated that the PNPs significantly enhanced the bioavailability of PTX-RESV PNPs than pure PAX and RESV. Thus, the study concluded that PAX-RESV PNPs combination could significantly enhance anti-glioma activity, and this could be developed into a potential glioma treatment strategy.

## 1. Introduction

Central nervous system tumors and malignancies showed 45% and 77% occurrence of gliomas, respectively [1]. Gliomas make up approximately 5% of the 5-year survival rate [2]. Radiotherapy, chemotherapy and resection are the common treatment strategy for gliomas [3], but these therapies were considered inefficient in managing cancer because the survival rate was less than 15 months [4]. This insufficiency to combat the disease was also due to the resistance and toxic effect of anti-neoplastic drugs.

Paclitaxel (PAX) is widely used in clinical trials for the treatment of various cancers such as lung, ovary, breast and glioma. The PAX is acting by stimulating the tubulin polymerization, which leads to G1 or G2/M cell cycle arrest and apoptosis [5]. PAX is also activating the production of reactive oxygen species (oxidative stress) found in many cancer cells including glioblastoma [6]. Owing to oxidative stress, the high mortality rate of cells could lead to the unknown molecular pathways of PAX resistance. Lately, the investigations have resulted in a PAX and resveratrol (RESV) combination that could reverse the multidrug resistance (MDR) on various cancer cell lines [7,8]. However, the poor blood–brain tumor barrier (BBTB) and blood–brain barrier (BBB) permeation of the active agents are the additional limiting factor for an effective glioma management [9]. Therefore, there is an urgent need to find a novel anticancer drug which has higher therapeutic index and efficiency for a glioma treatment [10].

The novel drug delivery systems are used to enhance the efficacy and increase the solubility of therapeutic agents [11]. Drug loaded nanoparticles are an efficient and non-invasive technique to treat cerebral diseases. Polymeric Nanoparticles (PNPs) are found to be the most likely drug nano carriers due to their excellent biocompatibility, targeting ability, enhanced residence time and convenient formulation technique [7,11].

Soluplus^®^ (polyvinyl caprolactam-polyvinyl acetate-poly-ethylene glycol) is a graft copolymer with a typical molecular weight (MW) ranging from 90,000 to 140,000 g/mol. Soluplus^®^ has a hydrophobic core that can hold lipophilic drug. The recent study has revealed that Soluplus^®^ and D-α-tocopheryl polyethylene glycol 1000 succinate (TPGS1000) nanoparticles were utilized to increase the BBB permeation [12]. Based on the previous investigations, an effective novel drug delivery system (NDDS) was planned for combination therapies (PAX and RESV) to deliver the two drugs instantaneously to the cancer sites to achieve synergistic anticancer activity [8,13].

The present study developed PAX-RESV loaded Soluplus PNPs by thin film hydration technique (Figure 1). The effect of the Soluplus, PAX-RESV and TPGS1000 concentrations on particle size, polydispersity index (PDI) and Percentage Entrapment Efficiency (%EE) of prepared nano-formulations was screened and optimized using experimental design (Box Behnken design (BBD)). The optimized PAX-RESV PNPs were further analyzed for several physicochemical evaluations, such as FT-IR, PXRD, SEM, and in vitro drug release studies were performed. Then, the individual and synergistic effect of pure PAX, pure RESV, PNPs (PAX and RESV loaded) were evaluated by in vitro cytotoxicity assay against C6 cells. Finally, the pharmacokinetic and bio distribution study was conducted to evaluate BBB permeation of prepared PNPs.

## 2. Materials and Methods

### 2.1. Materials

Soluplus and Resveratrol were obtained as a plentiful gift from BASF, Tifton, GA, USA. Paclitaxel was gifted by Shilpa Medicare Ltd., Bangalore, India. Dulbecco’s Modified Eagle’s medium (DMEM), Trypsin EDTA and Dulbecco’s phosphate buffer 7.4 pH, Penicillin-Streptomycin and glutaMAX^TM^ were purchased from Thermo Fisher Scientific (Pittsburgh, PA, USA). Di-methyl Sulfoxide (DMSO) molecular grade and 3-(4, 5-Dimethylthiazol-2-yl)-2, 5- diphenyl tetrazolium bromide) (MTT) were purchased from Sigma-Aldrich, St. Louis, MO, USA.

Animal glioma cell line (C6 ATCC^®^ CCL-107) was obtained from American Type Culture Collection (ATCC, Rockville, MD, USA). Cells were maintained in complete DMEM media, supplemented with 10% *v*/*v* FBS and 1% *v*/*v* Penicillin-Streptomycin and glutaMAXTM solution at 37 ± 0.5 °C with 5% carbon dioxide (CO2) in an incubator [14].

### 2.2. Reversed Phase-High-Performance Liquid Chromatography (RP-HPLC)

The RP-HPLC system (LC 2030C, Shimadzu, Japan) was equipped with a reverse phase Symmetry C18 column (3.5 μm, 4.6 × 75 mm^2^) and an ultraviolet (UV) detector.

#### 2.2.1. Mobile Phase Selection of PAX

The ratio of acetonitrile (CAN) and water (50:50) was used as a mobile phase at a flow rate of 1 mL/min for PAX determination. PAX was properly diluted with acetonitrile and injected directly into the RP-HPLC system using a run time of 10 min. The retention time (RT) was around 4.2 min with maximum absorption wavelength (λ max) of 227 nm. A series of standard PAX solutions were prepared using acetonitrile as diluent in the concentrations between 0.7–10 μg/mL.

#### 2.2.2. Mobile Phase Selection of RESV

The methanol, 10 mM potassium dihydrogen phosphate buffer (pH 6.8) and acetonitrile in the ratio of 60:30:10, *v*/*v*/*v* was considered as a mobile phase at a constant flow rate of 1 mL/min for determining RESV concentration. RP-HPLC was conducted in gradient elution, with the lamda max of 296 nm. The standard calibration curve was plotted using RESV (0.5–5 µg/mL) with a correlation coefficient of 0.999. The RT of trans-resveratrol was around 2.817 min [15].

### 2.3. Experimental Design

#### 2.3.1. Formulation and Optimization

Box Behnken design (BBD) was utilized for optimizing PAX-RESV PNPs using Design-Expert software (Version 12, Stat-Ease Inc., Minneapolis, MN, USA). The concentration of Soluplus (X1), 1:4 ratio of PAX: RESV (×2) and TPGS1000 (×3) (independent variables) on the basis of trial-and-error studies and % entrapment efficiency, PDI and particle size were selected as dependent variable. The independent and dependent variables are detailed in Table 1.

#### 2.3.2. Preparation of Functionalized Polymeric Nanoparticles

Thin film hydration method was used for the preparation of 1:4 ratio of PAX: RESV loaded PNPs [16]. In total, 17 formulations were prepared by BBD to recognize the Soluplus, PAX-RESV and TPGS1000 concentration effect. Initially, 1:4% *w*/*w* ratio of PAX and RESV (1 mg·mL^−1^) along with Soluplus in 10 mL of methanol was used for polymeric solution preparation, and the methanol present in the preparation is extracted using a rotary vacuum evaporator and obtained film was decked with TPGS1000. This solution was put at 1200 rpm for 30 min on magnetic stirring. The optimized PNPs were added with 5% *w*/*v* of mannitol (cryoprotectant), later the samples were frozen at −80 °C and lyophilized for a period of 48 h, and stored for further in vitro characterization. Polymeric loaded nanoparticles containing single drug (PAX and RESV) were also prepared using the above procedure with drug: Soluplus: TPGS1000 ratio being 1:30:1 mg·mL^−1^.

### 2.4. Characterization of Prepared PNPs

#### 2.4.1. Particle Size, PDI and Zeta Potential (ζ)

The mean particle size, PDI, and ζ of the formulated PNPs were evaluated through the DLS method using Malvern zetasizer [17].

#### 2.4.2. Percentage Entrapment Efficiency (%EE)

%EE of the prepared PAX and RESV PNPs was determined by adopting a centrifugation method. Briefly, 0.5 mL PAX-RESV loaded PNPs formulation were centrifuged at 14,000 rpm for 30 min at 4 °C in an ultracentrifuge. Then, 50 µL of supernatant was taken out by micropipette and diluted with 5 mL of HPLC grade ACN for estimating the amount of free PAX and RESV by the RP-HPLC method [17]. The %EE was calculated by using the following Equation (1):%EE = Amount of initial drug concentration − unentrapped drug in supernatant ÷ Amount of initial concentration × 100(1)

#### 2.4.3. Scanning Electron Microscopy (SEM)

Morphology of the PAX-RESV PNPs was determined using a scanning electron microscope (ZEISS, Oberkochen, Germany). PNPs were fixed on copper stubs; gold-coated and observed using an accelerating voltage of 10 kV [18].

#### 2.4.4. Fourier Transform Infrared (FT-IR) Spectroscopy

FT-IR (Shimadzu-8400S) (Kyoto, Japan) studies were done to ascertain drug–polymer compatibility. It was performed by using potassium bromide (KBr) pellet press technique. The pellets were prepared by using KBr with pure PAX, RESV, lyophilized Placebo, PAX NPs, RESV NPs and PAX-RESV PNPs (1:10 ratio) by applying a pressure of 4.5 tons. The spectrum evaluations were done within the range of 4500 cm^−1^ to 500 cm^−1^ [18].

#### 2.4.5. Powder X-Ray Diffraction (PXRD) Analysis

PXRD analysis (PROTO, Ontario, Canada) for pure PAX, RESV, lyophilized Placebo, PAX NPs, RESV NPs and PAX-RESV PNPs were analyzed to determine the physicochemical nature and external structural changes of the drug during the nanoparticle preparation. The diffraction pattern was studied using copper as radiation source, with scanning range and scan rate being 10–40° (2θ°) 0.6 sec at 2θ and a step size of 0.0200° at 2θ, respectively [19].

#### 2.4.6. In Vitro Release Studies

Release studies of pure PAX, RESV and selected formulation PAX-RESV PNPs were performed by using a 1 mL capacity dialysis membrane bag (12–14 kDa). The dialysis membrane was activated by soaking the membrane overnight in Phosphate Buffered Saline (PBS) with pH of 7.4 and then 1 mL of 0.5 mg·mL^−1^ concentration of pure drug and optimized PNPs were loaded into the dialysis membrane bag. This bag was placed beaker containing 100 mL of 0.2% Tween 80 PBS with continuous stirring at 100 rpm that was maintained at a temperature of 37 ± 0.5 °C. Aliquot (100 µL) of samples were withdrawn at various timepoints (0.1, 0.2, 0.3, 0.4, 0.5, 1, 2, 3, 4, 8, 12, 24, 36 h), and a net equal amount PBS was replaced to maintain constant release volume. The unknown concentrations of PAX and RESV were estimated by reverse phase HPLC technique (LC-2030C, Shimadzu, Japan) [17].

#### 2.4.7. In Vitro Cytotoxicity Studies

The stock solution of PAX (100 mmol), RESV (200 mmol) concentration were prepared by using DMSO. The required PAX, RESV solution used to treat C6 cell contained less than 0.1% DMSO. The lyophilized Placebo, lyophilized PAX NPs, lyophilized RESV NPs and lyophilized PAX-RESV PNPs were directly dissolved with cell culture medium.

Animal glial cells C6 was used to evaluate the cytotoxicity of individual and the combination of pure PAX, RESV, Placebo, and PAX and RESV PNPs. Cell survival rate was estimated by MTT assay. C6 (1.0 × 10^6^ cells/well) cells were dispersed in 100 μL of complete DMEM media for 24 h to attain complete confluence, these C6 cells were then exposed to different concentrations of pure PAX (0.39, 0.78, 1.56, 3.12, 6.25 and 12.5 μM), RESV (7.81, 15.62, 31.25, 62.5, 125 and 250 μM), Placebo, PAX, and RESV PNPs. Hundred micromolar cisplatin and 0.1% DMSO in DMEM were used as positive control (PC) and vehicle control (VC), respectively. Cells with complete media were denoted as the control, while wells without cells were considered as the negative control (NC). Microplates were nurtured in 5% carbon dioxide environment for 24, 48 and 72 h at a temperature of 37 °C. Later, 20 µL (10 mg·mL^−1^) of MTT solution was added to each microplate well as a reagent for identifying cell viability, after which the Optical Density (OD) was measured using a microplate reader (Thermo Scientific^®^ Varioskan Flash Multimode Reader, Winooski, VT, USA) at 570 nm. The mean OD for each set of wells were calculated [20,21]. The % cell viability was then calculated by using the below Equation (2).
% Cell viability = Mean OD (test-blank)/mean OD (control-blank) × 100%(2)

#### 2.4.8. In Vivo Pharmacokinetic and Brain Distribution Studies

For in vivo pharmacokinetic studies, all experimental animals were acclimatized to the laboratory conditions for a period of one week prior to the initiation of the experiment. The prepared study protocol approval was obtained by the Institutional Animal Ethics Committee (IAEC: 155/PO/Re/S/99/CPCSEA). Swiss albino weighing 20–25 g were selected for the study [22].

Swiss albino mice were injected intravenously (i.v.) via the tail vein with 5 mg/kg of PAX PNPs and 20 mg/kg of RESV PNPs. Mice were divided into three groups (3 mice for each time interval) e.g., group A (PAX), group B (RESV), and group C (PAX-RESV PNs). After the i.v. administration of the above-mentioned drugs, the blood sample was collected from the retro-orbital sinus at 0.05, 0.15, 0.3, 0.45, 1, 2, 4, 6 and 8 h, and 24 h time intervals. Then the blood samples were centrifuged at 4000 rpm for 15 min to separate the plasma. The plasma samples were stored at −80 °C until being analyzed for PAX and RESV content. At the end of the pharmacokinetic study, all the animals were anesthetized before being sacrificed. The brain tissues were excised and rinsed in PBS (pH 7.4) and stored in 10% formalin solution. PAX and RESV in the brain tissue was extracted with ethyl acetate (EA) at a ratio of 3:1 (EA:tissue), homogenized for 3 min and centrifuged at 8000 rpm for 3 min. Then, the supernatant was dried under nitrogen and re-dissolved with ACN, followed by RP-HPLC analysis. The elimination rate constant (ß), area under the curve (AUC), total body clearance (CL) = Dose/AUC, volume of distribution (Vdarca) = CL/ß and elimination half-life (t1/2ß) were calculated by the non-compartmental model using Pumas–Julia Computing software (Newton, MA, USA).

### 2.5. Statistical Analysis

Design-Expert software (Version 12, Stat-Ease Inc., and Minneapolis, MN, USA.) was utilized to obtain PAX-RESV PNPs. One-way analysis of variance (ANOVA) test was performed to compare differences among nano-formulations by utilizing Graph Pad software (San Diego, CA, USA). CompuSyn software version 1.0, (CompuSyn, Inc., New York, NY, USA; Cambridge, MA, USA) was used to obtain values for 50% inhibitory concentration (IC_50_) and 50% combination index (CI_50_) values.

## 3. Results and Discussion

The objective of this research is to develop PAX and RESV loaded Soluplus PNPs, then evaluate for various in vitro physicochemical parameters, and in vivo pharmacokinetic studies were performed to determine the BBB permeation using Swiss albino mice.

### 3.1. Formulation and Optimization of PAX and RESV PNPs

Soluplus was utilized as a carrier to encapsulate PAX and RESV and TPGS1000 was used to decorate the drug loaded nano carrier by employing thin film hydration technique. For the optimization of the prepared formulation, the BBD was used by keeping Soluplus, PAX-RESV and TPGS1000 ratio (independent factors), and particle size, PDI, % EE (dependent factors) were recorded in Table 2. Based on the cell line studies, the drug ratio (PAX: RESV) of 1:4 (*w*/*w*) was used in the optimized formulation.

### 3.2. Response Analysis of Prepared Formulations

#### 3.2.1. Particle Size of PAX-RESV PNPs

Particle size of prepared formulations ranged from 102.9 to 945.5, and ANOVA for quadratic model (Equation (3)) results in a F-value of 26.10 with *p* < 0.05 indicating the model being significant. The result concluded that the mean particle size was reduced significantly by increasing Soluplus concentration, whch might be due to the low critical micellar concentration (CMC) which may increase the formation of more uniform particles. However, the combination of Soluplus and TPGS_1000_ had an optimal effect around the midpoint concentration; this can be predicted by the fact that the Soluplus and TPGS_1000_ decreased the surface tension to reduce particle size. Further addition of polymer concentration more than the midpoint may cause the aggregation and incorporation of additional TPGS_1000_ molecules on the surface of the formulated PNPs. It was also observed that the particle size increases as enhanced insoluble drug concentration increases the CMC value of Soluplus which increases the particle size.
(3)$$\begin{array}{ll} \mathrm{Particle}\;\mathrm{Size}= & +\;1134.62029-56.28474\times \mathrm{Soluplus}+251.07038\times \mathrm{PAXRESV}\\ & -\;976.48338\times \mathrm{TPGS}-8.59126\times \mathrm{Soluplus}\times \mathrm{PAXRESV}\\ & \times \;+21.18899\times \mathrm{Soluplus}\times \mathrm{TPGS}+248.31290\times \mathrm{PAXRESV}\\ & \times \;\mathrm{TPGS}+0.796619\times {\mathrm{Soluplus}}^{2}+0.884873\times {\mathrm{PAXRESV}}^{2}\\ & +\;397.175\times \;\mathrm{TPGS}² \end{array}$$

#### 3.2.2. PDI of PAX-RESV PNPs

ANOVA for quadratic model (Equation (4)) resulted in a F-value of 71.68 (PAX-RESV PNPs) with *p*-value being less than <0.05, indicating model being statistically significant. PDI of prepared formulations are shown in Figure 2b. The changes in the PDI may be due to the drug–polymer interaction during the thin film formation. The optimum TPGS1000 concentration reduced the surface tension, resulting in uniform particles during hydration of the thin film. Conversely, high PAX-RESV or TPGS1000 concentration may have had an effect on the hydration phase, and therefore PNPs with wide particle size ranges were prepared, resulting in high PDI values.
PDI = +1.57440 − 0.034019 × Soluplus − 0.190328 × PAXRESV − 1.8154 × TPGS − 0.007491 × Soluplus × PAXRESV + 0.033453 Soluplus × TPGS × PAXRESV × TPGS + 0.000144 × Soluplus^2^ + 0.072943 × PAXRESV^2^ + 0.315945 × TPGS²(4)

#### 3.2.3. %EE of PAX-RESV PNPs

ANOVA for quadratic model (Equation (5)) showed a F-value of 59.93 with a *p*-value (≤0.05) which indicates that the analyzed model being significant. The initial drug concentration with maximum Soluplus had showed significant %EE, which may be due to enhanced solublization of PAX-RESV in Soluplus (Figure 2c). Further, increased drug concentration had decreased the %EE. It was also observed that the % EE had no effect on TPGS1000 concentration. Thus, it could be concluded that for the manufacture of PNPs with uniform size, PDI and %EE, the optimum levels of the polymer and drug need to be carefully chosen.
(5)$$\begin{array}{ll} \%\mathrm{EE}= & +\;56.76928-0.225088\times \mathrm{Soluplus}-35.39013\times \mathrm{PAXRESV}-103.80762\times \;\\ & \mathrm{TPGS}-0.157914\times \mathrm{Soluplus}\times \mathrm{PAXRESV}+0.893252\times \mathrm{Soluplus}\times \mathrm{TPGS}+\;\\ & 46.14570\mathrm{PAXRESV}\times \mathrm{TPGS}+0.051710\times {\mathrm{Soluplus}}^{2}+0.315945\times \;\\ & {\mathrm{PAXRESV}}^{2}+61.15048\times {\mathrm{TPGS}}^{2} \end{array}$$

#### 3.2.4. Particle Size, PDI, Zeta Potential, and %EE

Thin film hydration technique was used for the development of PAX-RESV PNPs. Soluplus was used to prepare the PNPs as it offers various advantages over other polymers. Soluplus is biocompatible and has been approved by US-FDA, also it is used clinically. The process was optimized to obtain the homogenous PNPs with maximum %EE. The particle size, PDI, and %EE of PAX-RESV PNPs was optimized by utilizing overlay plot as shown in Figure 2d. The particle size, PDI, Zeta potential, %EE of optimized PAX-RESV PNPs and Placebo were found to be 102.9 ± 0.17, and 77.8 ± 0.77 PDI of 0.257 ± 0.02, and 0.128 ± 0.011, Zeta potential of −2.83 ± 0.24, and −1.17 ± 0.41, %EE of 62.7 ± 2.3, 68.7 ± 3.2 and NA %, respectively. Besides, the individual drug loaded PNPs (PAX PNPs and RESV PNPs) were formulated for comparison as well. The particle size, PDI, Zeta potential, %EE of prepared PAX PNPs and RESV PNPs were found to be 106.5 ± 0.14, and 114.0 ± 0.21 PDI of 0.242 ± 0.03, and 0.257 ± 0.01, Zeta potential of −1.91 ± 0.14, and −3.11 ± 0.24, %EE of 60.2 ± 2.3, 62.4 ± 3.2, respectively. Figure 3 represents the particle size, and Zeta potential of prepared PAX PNPs, RESV PNPs, PAX-RESV PNPs and Placebo PNPs. The particle size of the optimized PAX-RESV PNPs was very near to the particle size of Doxil^®^ and Abraxane^®^, whose particle size was in the range of 130–150 nm [23]. Supporting the earlier results, the NPs of ~100 nm might passively target to cancer tissues through enhanced retention effect (EPR) with increased tumor permeability [24], causing a decrease in renal excretion [25], resulting in enhanced drug accumulation in cancer tissue. The %EE is a vital factor to determine the drug release from the carrier, as the results exhibited the highest %EE to which the PNPs may afford enough space for encapsulating the drugs and prevent them from leaking, and this parameter could offer noteworthy benefits in anticancer treatment [26].

### 3.3. Scanning Electron Microscopy (SEM)

Figure 4III shows the surface morphology of the optimized PAX-RESV PNPs were evaluated by using SEM, which resulted in spherical shaped particles with smooth surfaces and homogeneity.

### 3.4. Fourier Transform Infrared (FT-IR) Spectroscopy

The FT-IR of Pure PAX, RESV, Placebo, PAX-RESV PNPs, PAX, and RESV PNPs was shown in Figure 4I(A–F) and their corresponding characteristic peak positions are listed in Table 3. The data obtained confirms the presence of this characteristic’s peaks in PAX PNPs, RESV PNPs and PAX-RESV PNPs (Figure 4I(D–F)) matching to the peaks of the pure drugs (Figure 4I(A,B)) without any major shifts. Thus, FT-IR studies concluded polymer drug compatibility.

### 3.5. Powder X-ray Diffraction (PXRD) Analysis

Figure 4II shows PXRD patterns of Pure PAX, RESV, Placebo, PAX-RESV PNPs, PAX, and RESV PNPs prepared by the thin film hydration method. Pure PAX and RESV exhibited the main characteristic PXRD peaks at 2θ = 16.98°, 17.90°, 20.37°, 21.08°, 22.02°, 24.06°, 25.25°, 27.86°, 34.99°, 36.13° and 11.02°,13.73°, 17.31°, 17.97°, 19.88°, 21.36°, 23.17°, 25.57°, 27.21°, 30.36°, 31.37°, 33.58°, 36.27°. After the thin film hydration process, nearly all the main characteristic peaks of crystalline of PAX and RESV became weak, especially peaks at 2θ = 21.08°, 22.02°, 24.06 and 21.36°, 23.17°, 25.57°, 27.21° in the diffractograms of PAX PNPs, RESV PNPs, and PAX/RESV PNPs because the two drugs existed as amorphous form or in molecular state [27]. This indicated that the thin film hydration process has manipulated the crystallinity of PAX and RESV by encapsulating the drug molecule inside the Soluplus and TPGS1000. The results of PXRD supported the fact that the drug was entrapped inside the PNPs.

### 3.6. In Vitro Release Study

The in vitro drug release study results were revealed that the pure PAX and RESV release from the dialysis bag was more when compared to the PAX-RESV PNPs. Figure 5 displayed biphasic release pattern with a burst drug release was detected from the PAX-RESV PNPs initially, which may be due to unentrapped drug particle. The diffusion of free PAX and RESV was found within 3 h of the study, while only 25% of PAX-RESV was released from PNPs at the same time. About 60–70% of PAX and RESV were released from PNPs at 12 h and the prolonged drug release were seen over a period of 36 h.

### 3.7. Cell Cytotoxicity Study

MTT based cytotoxic effect of the developed PAX, RESV, PAX PNPs, RESV PNPs and PAX-RESV combined PNPs were evaluated against C6 cells. Initially, the cells were treated with free PAX, RESV, PAX and RESV PNPs for dose and time dependent responses as shown in Figure 6. Later, PAX-RESV combined PNPs at equivalent concentration and the cytotoxicity was measured (Figure 7). The results of cytotoxicity were compared with free PAX, RESV and untreated cell designated as control. The obtained results, IC50 value for C6 cells obtained for free PAX was at 24 h (4.588 ± 0.19 μM), 48 h (2.97 ± 0.14 μM) and 72 h (1.49 ± 0.21 μM), while IC50 value of PAX PNPs was at 24 h (2.329 ± 0.12 μM), 48 h (1.59 ± 0.09 μM) and 72 h (0.83 ± 0.01 μM). The IC50 value for free RESV was found at 24 h (89.85 ± 1.29 μM), 48 h (49.15 ± 0.94 μM) and 72 h (44.57 ± 2.01 μM), while IC50 value of RESV PNPs was at 24 h (59.62 ± 1.01 μM), 48 h (26.71 ± 1.24 μM) and 72 h (24.37 ± 0.11 μM). The IC50 values of PAX, RESV, PAX PNPs and RESV PNPs results suggested that the drug loaded PNPs increased two-fold than pure drugs against C6 cell lines. The PNPs Placebo and VC (DMSO) do not cause any cytotoxicity and were nontoxic to the cells, revealing their biocompatibility. In addition, mannitol that was used in all the prepared formulation, including placebo, did not show any toxic effect on C6 cells.

#### Combination Effect

As shown in Figure 6, the IC50 value of free PAX and RESV at 24 h against C6 (5 and 90 μM) cell lines were taken for a combinatory effect study at 48 h of incubation. As expected, the combination of PAX and RESV treatment on C6 cells exhibited the synergistic effect (CI < 0.83) (Figure 7A). Besides, PAX-RESV-loaded PNPs exhibited the highest cytotoxicity on C6 cells (Figure 7B).

### 3.8. Pharmacokinetic Studies

The pharmacokinetic parameters of free PAX, RESV, and PAX-RESV PNPs were determined by i.v. administration of the above-mentioned formulations to different groups of Swiss albino mice, and normal saline was used as a control. The obtained pharmacokinetic data are presented in Table 4 and Figure 8A. PAX and RESV PNPs exhibited a slow clearance rate from the blood even after 8 h of administration. PAX and RESV solution showed rapid initial clearance rate from the blood followed by a decrease in clearance after 2 and 4 h of administration, respectively. The significant pharmacokinetics parameters were observed between PAX and RESV solution and PAX-RESV PNPs exhibited a decrease in the T1/2 along with an increase in mean residence time (MRT). The area under curve (AUC 0–24) and AUC 0–∞ were lower for PAX-RESV PNPs, clearly indicating that lower concentration of free PAX and RESV in plasma. Further, plasma clearance and elimination rate constant was significantly higher for PAX-RESV PNPs than for free PAX and RESV. These results clearly suggested that the PAX-RESV PNPs do not release drugs in plasma and have the ability to reach the target site.

The brain distribution of PAX, RESV solution, PAX and RESV loaded PNPs was quantitatively assessed in mice after i.v. administration. As shown in Figure 8B, i.v. administered PAX and RESV loaded PNPs had the highest drug concentration in the brain after dosing. This could be due to faster and better absorption across BBB (43.57 ± 3.34 and 63.25 ± 4.74 ng/mL), and can be attributed to inhibition of glycoprotein produced by BBB by TPGS1000 [28], resulting in decreased outflow of the NPs from the tumor cells. The free PAX and RESV solution (12.4 ± 0.84 and 23.14 ± 1.32 ng/mL) administered i.v. showed significantly lesser brain concentration as compared to drug loaded PNPs. The results exhibited that the concentration of pure drug present in brain was lesser than the optimized PNPs formulation. A four-fold increase in the concentration of PNPs formulation was observed in comparison to pure drug and control-treated mice.

## 4. Conclusions

The present study developed PAX-RESV PNPs successfully with anti-glioma potential. The results indicated that PNPs had good drug entrapment efficiency and appropriate particle size, along with prolonged in vitro drug release. PAX-RESV-loaded Soluplus PNPs showed better cytotoxicity against glioma cells in comparison to free PAX, RESV, PAX PNPs and RESV PNPs. It can be concluded that PAX-RESV combined PNPs could be utilized for effective therapeutic management of glioma with reduced toxicity.

## Figures and Tables

**Figure 1 polymers-13-03210-f001:**
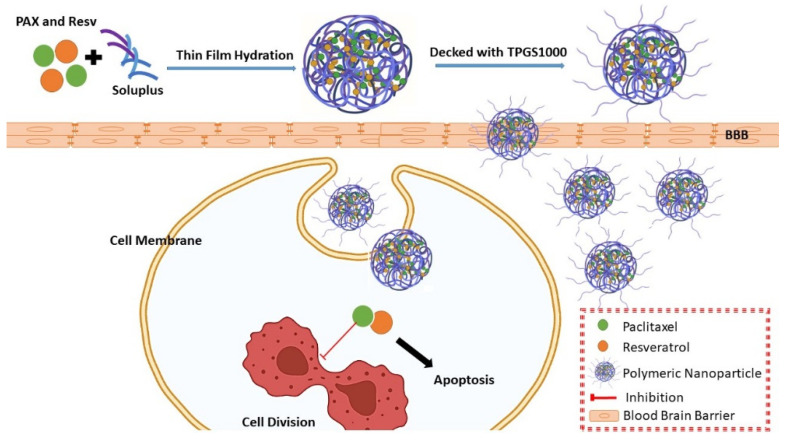
Schematic representation of the PAX-RESV loaded PNPs for anti-glioma activity.

**Figure 2 polymers-13-03210-f002:**
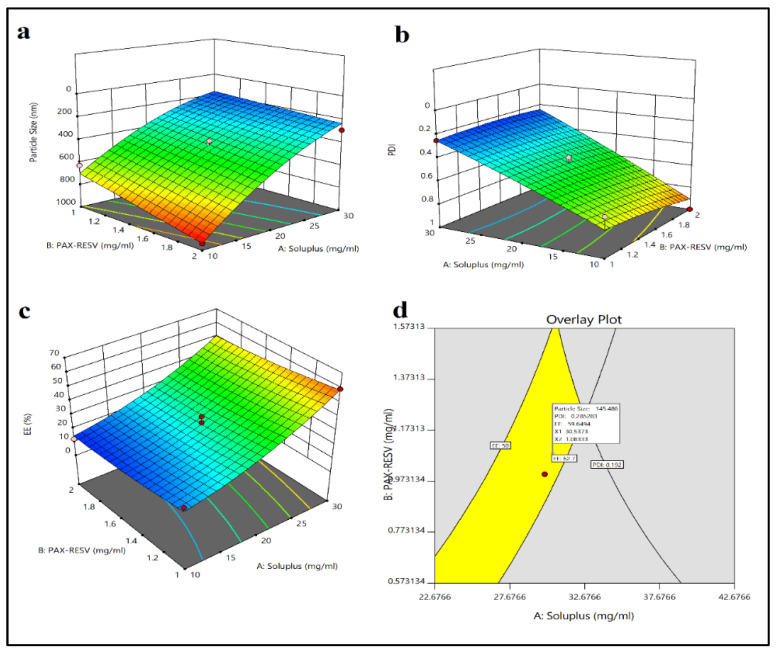
3D plots depicting the impact of Soluplus, PAX-RESV and TPGS1000 ratio (independent factors) on particle size, PDI, % EE (dependent factors) of the prepared PAX-RESV PNPs (**a**–**c**) and Overlay plot of optimized PAX-RESV PNPs (**d**).

**Figure 3 polymers-13-03210-f003:**
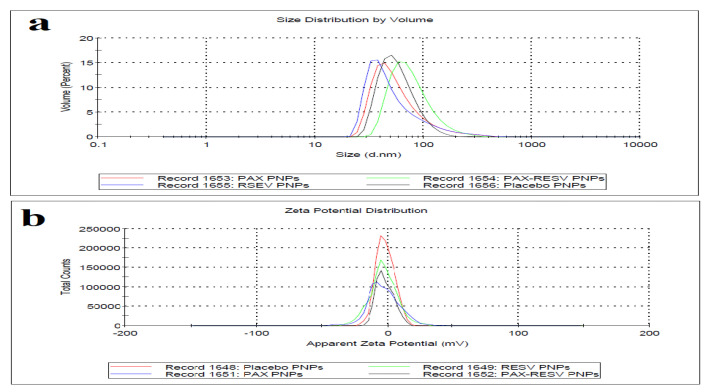
(**a**) Particle size distribution of optimized PAX-RESV PNPs (green line), RESV PNPs (blue line), Placebo (black line) and PAX PNPs (red line); (**b**) Zeta potential of optimized Placebo (red line), RESV PNPs (green line), PAX PNPs (blue line) and PAX-RESV PNPs (black line).

**Figure 4 polymers-13-03210-f004:**
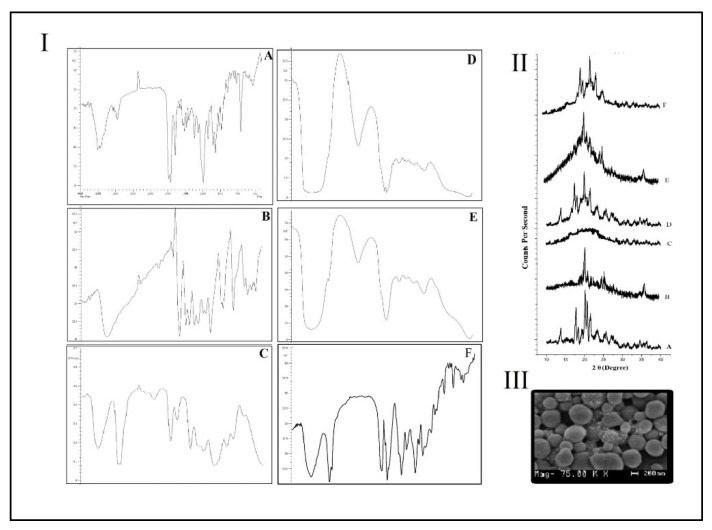
(**I**) FT-IR Spectrum of (**A**) PAX; (**B**). RESV; (**C**) Placebo; (**D**) PAX PNPs; (**E**) RESV PNPs; (**F**) PAX-RESV PNPs: (**II**) PXRD Analysis of (**A**) PAX; (**B**) RESV; (**C**) Placebo; (**D**) PAX PNPs; (**E**) RESV PNPs; (**F**) PAX-RESV PNPs: (**III**) The SEM images analysis of PAX-RESV PNPs.

**Figure 5 polymers-13-03210-f005:**
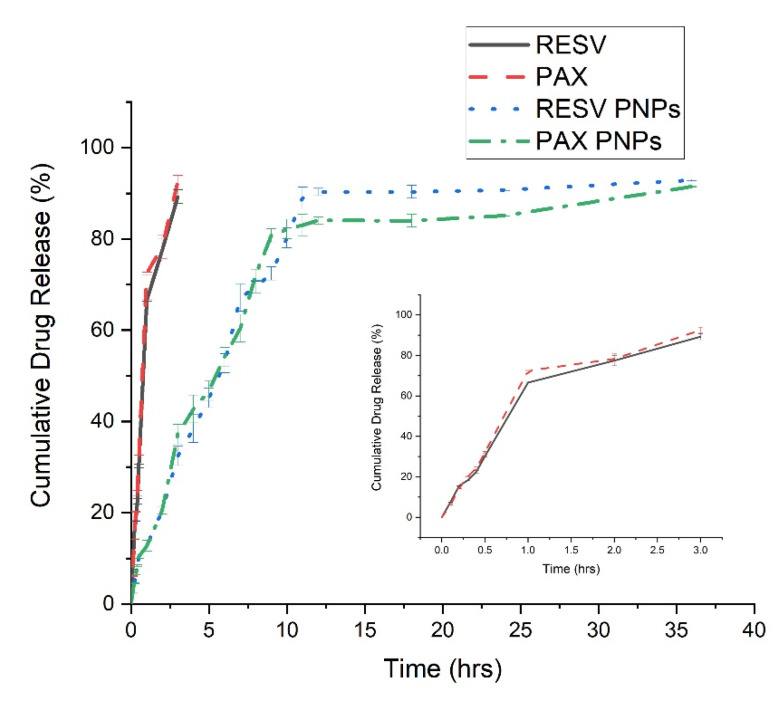
In vitro drug release profile of free PAX, RESV, PAX and RESV PNPs.

**Figure 6 polymers-13-03210-f006:**
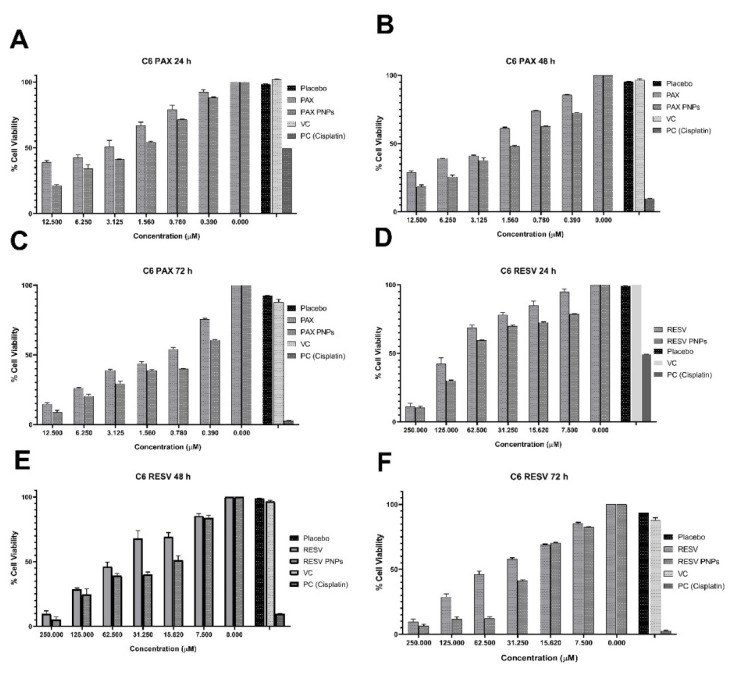
In vitro Cytotoxicity of free PAX and PAX PNPs (**A**) (24 h), (**B**) (48 h), and (**C**) (72 h) of C6 cell lines. In vitro Cytotoxicity of free RESV, and RESV PNPs (**D**) (24 h), (**E**) (48 h), and (**F**) (72 h) on C6 cell lines. Data represent the mean± standard deviation (*n* = 3).

**Figure 7 polymers-13-03210-f007:**
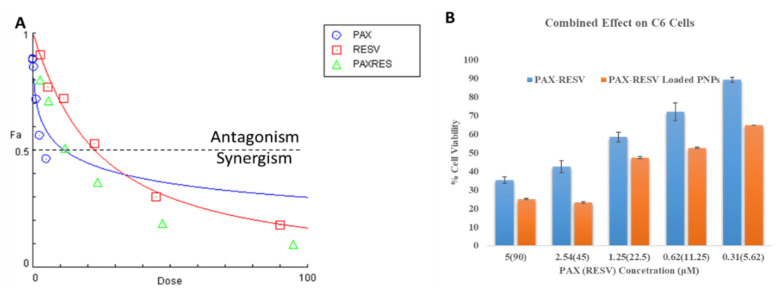
In vitro Cytotoxicity of Combined free PAX/RESV and PAX/RESV PNPs. (**A**) Combined dose effect of free PAX/RESV on C6 cells; (**B**) % Cell Viability of PAX/RESV PNPs on C6 cells.

**Figure 8 polymers-13-03210-f008:**
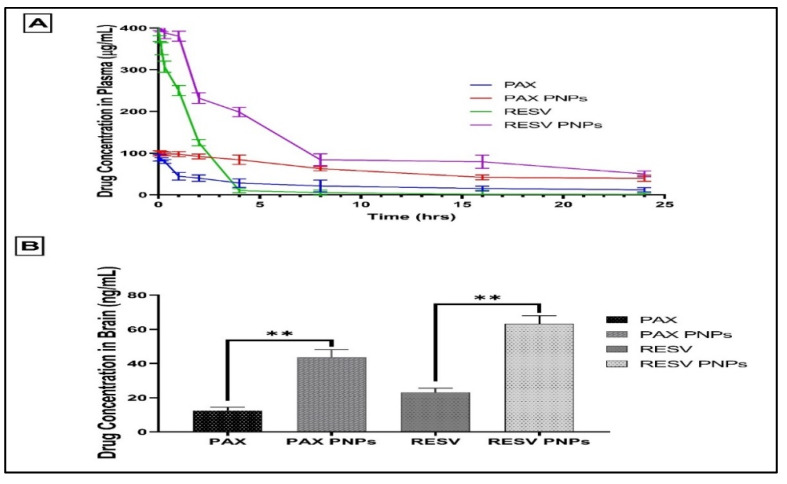
In vivo pharmacokinetics and brain distribution of PAX-RESV loaded PNPs or free PAX and RESV. (**A**) The In vivo pharmacokinetics results of PAX-RESV loaded PNPs or free PAX and RESV concentration in Plasma. (**B**) The In vivo brain distribution results of PAX-RESV loaded PNPs or free PAX and RESV concentration in Brian. ** indicates *p* < 0.01 results are statistically significant. Data represent the mean± standard deviation (*n* = 3).

**Table 1 polymers-13-03210-t001:** Variables in Box Behnken design for preparation and optimization of PAX and RESV PNPs.

Factors	Levels	
Independent variable	Low	High
X1 = Soluplus (mg·mL^−1^)	10	30
X2 = PAX-RESV (mg·mL^−1^)	1	2.0
X3 = TPGS1000 (% *w*/*v*)	0.1	0.5
Dependent variable	Goals	
Y1 = Particle size (nm)	Decrease	
Y2 = PDI	Decrease	
Y3 = Entrapment Efficiency (%)	Increase	

**Table 2 polymers-13-03210-t002:** BBB designs factors and Observed responses for PAX-RESV PNPs.

Std	Run	X1	X2	X3	Y1	Y2	Y3
9	1	20	1	0.1	441.3	0.864	31.7
1	2	10	1	0.3	627.1	0.663	22.4
7	3	10	1.5	0.5	841.2	0.751	18.4
11	4	20	1	0.5	383.1	0.334	35.5
6	5	30	1	0.1	102.9	0.192	62.7
12	6	20	2	0.5	537.6	0.384	42.1
3	7	10	2	0.3	945.5	1	12.4
2	8	30	1	0.3	197.2	0.254	59.2
8	9	30	1.5	0.5	228.3	0.253	60.4
16	10	20	1.5	0.3	421.7	0.548	28.7
5	11	10	1.5	0.1	895.5	0.872	17.7
10	12	20	2	0.1	447.4	0.542	22.5
15	13	20	1.5	0.3	411.2	0.457	25.4
13	14	20	1.5	0.3	472.7	0.474	30.4
17	15	20	1.5	0.3	398.7	0.458	34.7
4	16	30	2	0.3	294.7	0.315	48.7
14	17	20	1.5	0.3	421.7	0.548	28.7

X1 = Soluplus (mg·mL^−1^), X2 = PAX-RESV (mg·mL^−1^), X3 = TPGS1000 (% *w*/*v*), Y1 = Particle size (nm), Y2 = PDI and Y3 = Entrapment Efficiency (%).

**Table 3 polymers-13-03210-t003:** FT-IR Spectrum of Pure PAX, RESV, Placebo, PAX, RESV PNPs and PAX-RESV PNPs.

Peak Position					
PAX	PAX PNPs	RESV	RESV PNPs	PAX-RESV PNPs	Inter-Atomic Bond
3504.77	3512.49	3602.29	3674.52	3464.27	O-H Stretching vibration of phenol (Free)
2945.40	2951.19	2899.68	3030.27	2883.68	C-H Stretching (alkane)
2359.02	2368.66	2240.52	2291.51	2393.74	S-H Stretching
1529.60	1649.19	1638.14	1695.49	1633.76	N-H Bending
1375.29	1230.43	1224.30	1147.68	1373.36	C-O Stretching (alcohols, phenols)
765.77	824.91	843.50	823.63	842.92	C-H Bending (aromatic)
1253.77	1253.90	920.08	979.87	1242.20	C = C aromatic stretch
-	-	1432.54	1452.45	1438.94	O-H bending of phenols
1182.40	1034.221	1233.02	1234.48	1132.25	C-CO-C stretch and bending in ketone

**Table 4 polymers-13-03210-t004:** Pharmacokinetic parameters following i.v. administration.

Pharmacokinetic Parameters	Tissue/ Organ	PAX	PAX PNPs	RESV	RESV PNPs
C_max_ (µg/mL)	Plasma	88.75 ± 7.98	99.42 ± 5.98	374.5 ± 6.98	399.24 ± 4.18
T_max_ (min)		5	5	5	5
AUC_0–24_ (ng/mL)		324.1 ± 23	924.4 ± 79	352.7 ± 88	1241.5 ± 108
AUC_0–__∞_ (ng/mL)		533.8 ± 56	1409 ± 74	688.7 ± 34	286 2 ± 73
T_1/2_ (h)		4.68	12.34	2.12	8.21
K_E_ (h^−1^)		0.21	1.42	0.12	1.47
MRT_(0–t)_ (h)		1.24 ± 1.31	8.24 ± 3.22	0.3 ± 1.7	4.8 ± 2.4

C_max_: Maximum Concentration; T_max_: Time to reach maximum concentration; AUC Area under curve; T_1/2_: Half-life; KE: Elimination rate; MRT: Mean residence time.

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
