# Peer review of "Fabrication and Characterization of Paclitaxel and Resveratrol Loaded Soluplus Polymeric Nanoparticles for Improved BBB Penetration for Glioma Management"

_polymers, 2021, doi:10.3390/polym13193210_

Round 1

Reviewer 1 Report

The authors have designed a soluplus system that can enter the brain to deliver Resv or paclitaxel. The article overall is well written and very interesting but there are several things that must be explained or more completely analyzed during the comparisons.

minor changes:

      1) general english corrections

       2)  polymer weight from 90,000 to 1,40,000    is mistyped

      3) methodology for HPLC of resv gradient not described

      4 statisics missing in figure 6..... did combination therapy truly lead to an increase?

Major changes

     The authors state that 80% the drugs are released over 36 hours but this is very misleading as a cfast release of ~80 is seen in the first 10 hours followed by a minor release long term.

       The authors continue to cite the combined therapies of RESV-PAC PNPs where the two drugs are coincapsulated: however the authors only used this sample once for cell toxicity. Every other use was PNPs with one or the other drug. Release kinetics and BBB targeting/pharmicokinetics could GREATLY change when co-encapsulating drugs into the same NPs.

       Mannitol is used as a stabilizer in the NP formulation but mannitol is well known in the literature to disrubt cell membranes and to have effects on the BBB. The authors never discuss this. This would also lead to necessary controls where the Pax or Resv controls also contained mannitol that could lead to higher uptake and higher toxicity. 

Author Response

Reviewer 1 comments:

First of all, we appreciate the time spend by the honorable reviewers from their busy schedule to improve the quality of our MS. All the suggestions given by the reviewer have been addressed and highlighted in Yellow with track changes on.

Minor changes:

  1. general english corrections
  • We appreciate the concern of the honorable reviewer; the MS has been thoroughly revised for English corrections.
  1. polymer weight from 90,000 to 1,40,000 is mistyped
  • The error has been corrected and explained in the revised MS.
  1. methodology for HPLC of resv gradient not described
  • The RP-HPLC method for the determination of RESV was carried out with fixed mobile phase ratio of methanol, 10 mM potassium dihydrogen phosphate buffer (pH 6.8) and acetonitrile (60:30:10, v/v/v).
  1. statisics missing in figure 6..... did combination therapy truly lead to an increase?
  • We appreciate the concern of the honorable reviewer, in our article combinatorial effect details were observed in 3.7.1. section. Here, the individual cytotoxicity of PAX and RESV with respect to the combination of PAX-RESV showed synergistic effect which was revealed through isobologram analysis (Figure 7A). The section is highlighted for the reference.

Major changes:

  1. The authors state that 80% the drugs are released over 36 hours but this is very misleading as a cfast release of ~80 is seen in the first 10 hours followed by a minor release long term.
  • We appreciate the concern of the honorable reviewer; the statement has been duly modified in the revised MS to avoid discrepancies.
  1. The authors continue to cite the combined therapies of RESV-PAC PNPs where the two drugs are coincapsulated: however the authors only used this sample once for cell toxicity. Every other use was PNPs with one or the other drug. Release kinetics and BBB targeting/pharmicokinetics could GREATLY change when co-encapsulating drugs into the same NPs.
  • We appreciate the concern of the honorable reviewer; most humbly we will like to mention that we have used RESV-PAX co-loaded PNPs for all in-vitro and in-vivo analysis including drug release and bio-distribution/pharmacokinetics study, and individual PNPs were used for all assays except drug release and bio-distribution/pharmacokinetics study. In fact, the graphs are based on HPLC, when we perform HPLC of RESV-PAX co-loaded PNPs, we get two different peaks and on the basis of these peaks Figure 5 and 8 have been plotted. It means that RESV-PAX co-loaded PNPs formulation have given two different peaks. Thus, the individual drug loaded PNPs mentioned in the graph were on the basis of results of RESV-PAX co-loaded PNPs and not on the basis of individual PNPs analysis. Hope this clarify the doubts of the honorable reviewer. Some of following articles can be used as a reference for the same:
  • Shen Y, TanTai J. Co-Delivery Anticancer Drug Nanoparticles for Synergistic Therapy Against Lung Cancer Cells. Drug Des Devel Ther. 2020;14:4503-4510.
  • Karki, N., Tiwari, H., Tewari, C., Rana, A., Pandey, N., Basak, S., & Sahoo, N. G. (2020). Functionalized graphene oxide as a vehicle for targeted drug delivery and bioimaging applications. Journal of Materials Chemistry B, 8(36), 8116-8148.
  • Faivre L, Gomo C, Mir O, Taieb F, Schoemann-Thomas A, Ropert S, Vidal M, Dusser D, Dauphin A, Goldwasser F, Blanchet B. A simple HPLC-UV method for the simultaneous quantification of gefitinib and erlotinib in human plasma. J Chromatogr B Analyt Technol Biomed Life Sci. 2011 Aug 1;879(23):2345-50.
  • Kurangi, B., Jalalpure, S., & Jagwani, S. (2019). A validated stability-indicating HPLC method for simultaneous estimation of resveratrol and piperine in cubosome and human plasma. Journal of Chromatography B, 1122, 39-48.

  1. Mannitol is used as a stabilizer in the NP formulation but mannitol is well known in the literature to disrubt cell membranes and to have effects on the BBB. The authors never discuss this. This would also lead to necessary controls where the Pax or Resv controls also contained mannitol that could lead to higher uptake and higher toxicity. 
  • We appreciate the concern of the honorable reviewer; we agree with the reviewer that mannitol has effects on the blood brain barrier. However, it is also a fact that mannitol has been widely applied for different neurotherapy based researches. Some of the references are:
  • Super-selective Intra-arterial Repeated Infusion of Cetuximab for the Treatment of Newly Diagnosed Glioblastoma. https://clinicaltrials.gov/ct2/show/NCT02861898
  • Boockvar, J. A., Tsiouris, A. J., Hofstetter, C. P., Kovanlikaya, I., Fralin,. S., and Kesavabhotla, K. (2011). Safety and maximum tolerated dose of superselective intraarterial cerebral infusion of bevacizumab after osmotic blood-brain barrier disruption for recurrent malignant glioma. J. Neurosurg. 114, 624–632. doi: 10.3171/2010.9.JNS101223
  • E. Park, B. Singh, H.S. Li, J.Y. Lee, S.K. Kang, Y.J. Choi, et al. (2015) Enhanced BBB permeability of osmotically active poly(mannitol-co-PEI) modified with rabies virus glycoprotein via selective stimulation of caveolar endocytosis for RNAi therapeutics in Alzheimer׳s disease. Biomaterials, 38 (2015), pp. 61-71
  • Foley C.P., Rubin D.G., Santillan A., Sondhi D., Dyke J.P., Gobin Y.P., Crystal R.G., Ballon D.J. Intra-arterial delivery of AAV vectors to the mouse brain after mannitol mediated blood brain barrier disruption. J. Control. Release. 2014;196:71–78. doi: 10.1016/j.jconrel.2014.09.018.
  • Chen KB, Wei VC, Yen LF, Poon KS, Liu YC, Cheng KS, Chang CS, Lai TW. Intravenous mannitol does not increase blood-brain barrier permeability to inert dyes in the adult rat forebrain. Neuroreport. 2013 Apr 17;24(6):303-7.
  • Steiniger SC, Kreuter J, Khalansky AS, Skidan IN, Bobruskin AI, Smirnova ZS, Severin SE, Uhl R, Kock M, Geiger KD, Gelperina SE. Chemotherapy of glioblastoma in rats using doxorubicin-loaded nanoparticles. Int J Cancer. 2004 May 1;109(5):759-67.

Most humbly we will like to state that mannitol was used in all the prepared formulation including placebo, but we haven’t seen any toxic effect of placebo on C6 cells. We have added the same in the results and discussion section of the revised MS.

Reviewer 2 Report

The article titled “Fabrication and characterization of paclitaxel and resveratrol loaded soluplus polymeric nanoparticles for improved BBB penetration for glioma management” were the authors presented the preparation of PAX-RESV loaded polymeric nanoparticle and their characterization. They have also presented the individual and synergistic effects of pure PAX, pure RESV and the PNPs loaded with both the PAX and RESV. They have also investigated the BBB permeation of the loaded PNPs. Since, the future trend is to use single nanoparticle to deliver multiple drugs at a time and that too across BBB, hence this study is a very timely one. However, in my opinion a few of the concerns listed below needs visitation.

  1. The English language needs improvement throughout the manuscript to avoid the minor mistakes made, especially in terms of grammatical mistakes.
  2. Equation (1) needs to be rewritten following the mathematical rules properly, use parentheses.
  3. Line No. 171: Present the number of cells in a correct way.
  4. The units of concentration throughout the manuscript needs to be presented in a standard way (mg.mL-1 instead of mg/mL). Present all other units in a standard format, too.
  5. Can the phrase “BBB permeation” be better to use instead of “BBB penetration”?
  6. Provide a schematic presentation of the drug loaded polymeric nanoparticle designed in this study to help readers easily understand the carrier. This can be inserted at the last paragraph of the introduction section.
  7. In all the tables 1 to 4, remove the shades and prepare the tables in the standard format. Remove all the horizontal lines except the two at the top and one at the bottom.
  8. For Table 2, insert the information about X and Y terms at the bottom of the table to help reader understand what they represent.

Author Response

Reviewer 2 comments:

First of all, we appreciate the time spend by the honorable reviewers from their busy schedule to improve the quality of our MS. All the suggestions given by the reviewer have been addressed and highlighted in Yellow with track changes on.

  1. The English language needs improvement throughout the manuscript to avoid the minor mistakes made, especially in terms of grammatical mistakes.

  • As per the suggestion of the honorable reviewer, we have duly modified the MS to avoid grammatical errors.

  1. Equation (1) needs to be rewritten following the mathematical rules properly, use parentheses.

  • As per the suggestion of the honorable reviewer, we have duly modified the equation 1 in the revised MS.

  1. Line No. 171: Present the number of cells in a correct way.

  • The number of cells have been represented in the correct way in the revision.

  1. The units of concentration throughout the manuscript needs to be presented in a standard way (mg.mL-1 instead of mg/mL). Present all other units in a standard format, too.

  • Units throughout the MS has been duly modified as per the standard format and uniformity in the revised MS.

  1. Can the phrase “BBB permeation” be better to use instead of “BBB penetration”?

  • As per the suggestion of the honorable reviewer, we have duly modified the MS to change BBB penetration to permeation.

  1. Provide a schematic presentation of the drug loaded polymeric nanoparticle designed in this study to help readers easily understand the carrier. This can be inserted at the last paragraph of the introduction section.

  • As per the suggestion of the honorable reviewer, we have duly added Figure 1 as a schematic representation in the introduction section.

  1. In all the tables 1 to 4, remove the shades and prepare the tables in the standard format. Remove all the horizontal lines except the two at the top and one at the bottom.

  • The tables have been duly modified in the revised MS as per the suggested comment.

  1. For Table 2, insert the information about X and Y terms at the bottom of the table to help reader understand what they represent.

  • In Table 2, information about X and Y terms has been inserted in at the bottom of the table as suggested by the honorable reviewer.

Round 2

Reviewer 1 Report

I would like to thank the authors for their good work and clarification on the issues presented.

Reviewer 2 Report

The manuscript has been satisfactorily improved and may be accepted for publication.